# Discovery of 9*O*-Substituted Palmatine Derivatives as a New Class of Anti-COL1A1 Agents Via Repressing TGF-β1/Smads and JAK1/STAT3 Pathways

**DOI:** 10.3390/molecules25040773

**Published:** 2020-02-11

**Authors:** Tianyun Fan, Maoxu Ge, Zhihao Guo, Hongwei He, Na Zhang, Yinghong Li, Danqing Song

**Affiliations:** Institute of Medicinal Biotechnology, Chinese Academy of Medical Sciences and Peking Union Medical College, Beijing 100050, China; fty1668@163.com (T.F.); gemaoxu@outlook.com (M.G.); guozhihao96@163.com (Z.G.); hehwei@imb.pumc.edu.cn (H.H.); songdanqingsdq@hotmail.com (D.S.)

**Keywords:** hepatic fibrosis, palmatine, structure−activity relationship, COL1A1, TGF-β1/smads pathway, JAK1/STAT3 pathway

## Abstract

Twenty 9*O*-substituted palmatine derivatives were prepared and tested for their biological effect against collagen α1 (I) (COL1A1) promotor in human hepatic stellate LX-2 cells. The structure−activity relationship (SAR) indicated that the introduction of a benzyl motif on the 9*O* atom was favorable for activity. Among them, compound **6c** provided the highest inhibitory effect against COL1A1 with an IC_50_ value of 3.98 μM, and it also dose-dependently inhibited the expression of fibrogenic COL1A1, α-soomth muscle actin (α-SMA), matrix metalloprotein 2 (MMP2) in both mRNA and protein levels, indicating extensive inhibitory activity against fibrogenesis. A further primary mechanism study indicated that it might repress the hepatic fibrogenesis via inhibiting both canonical transforming growth factor-beta 1 (TGF-β1)/Smads and non-canonical janus-activated kinase 1 (JAK1)/singal transducer and activator of transcription 3 (STAT3) signaling pathways. Additionally, **6c** owned a high safety profile with the LD_50_ value of over 1000 mg·kg^−1^ in mice. These results identified palmatine derivatives as a novel class of anti-fibrogenic agents, and provided powerful information for further structure optimization.

## 1. Introduction

Liver injury induces inflammation, the wound-healing response, as well as the accumulation of extracellular matrix (ECM) proteins, followed by a process of hepatocyte regeneration to replace dead hepatocytes and restore the physiological liver mass, which explains the generation of hepatic fibrosis (HF) [1,2]. HF typically reverts after the elimination of the causative injury. However, HF can progress to irreversible and even fatal cirrhosis if the damage persists and a chronic response is established [3]. Nonalcoholic steatohepatitis (NASH), alcoholic liver disease, and viral hepatitis represent the major etiologies leading to HF [4]. Besides the epidemiological relevance, HF and, hence, cirrhosis also impose a considerable economic burden on society [4]. In the last decades, an immense effort has been undertaken to elucidate the mechanisms of HF and develop therapeutic approaches. However, no drug has yet been approved for routine clinical use [5]. Therefore, novel anti-fibrotic treatment is still in great need.

The proliferation and activation of hepatic stellate cells (HSCs), which might be induced by oxidative stress or multiple of chemokines, for example, transforming growth factor-beta (TGF-β) during liver injury, is the major cellular source of HF [6,7]. The production of excessive ECM, whose main component is collagen type I (COL1), is a hallmark of activated stellate cells [8]. In addition, fibrotic progression is characterized by the replacement of normal basement membrane (named as collagen type IV) with scar forming COL1 [9]. The content of COL1 increases remarkably during HF [9], and downregulating the expression of the COL1 might directly inhibit HF, and COL1 might be an effective target for an anti-HF study [8]. In view of the key role of COL1 in promoting hepatic fibrosis, an in vitro cell model that was based on the COL1A1 promoter was established, in which the activity of luciferase could be elevated by activators, like TGF-β1, or inhibited by candidate agents [10]. This model had been successfully applied to the screening and evaluation of anti-hepatic fibrosis drug candidates in our earlier studies [11,12,13,14]. The compound stood out in the anti-COL1A1 assay might effectively repress HF in vivo [11].

Our group has been dedicated to the structure modification and biological activity discovery of protoberberine alkaloid, and successfully constructed a protoberberine derivatives library. Their anti-bacterial, anti-viral, and anti-inflammatory activities were discovered [15,16,17,18,19]. Therefore, in our latest study, the COL1A1-based high-throughput screening model screened the palmatine derviatives, a very important family of the protoberberine alkaloid, for the very first time. We were lucky enough to find 9-bezoyl palmatine (**1**) as an ideal lead compound with a 24.73% inhibitory rate against COL1A1 at the concentration of 20 μM. Since the application of palmatine (**2**) and its derivatives as anti-HF agents had never been discovered, it might be interesting to conduct an anti-COL1A1 structure-activity relationship (SAR) study, and develop and discover new anti-fibrogenic agents of its kind.

In the present study, taking **1** as the lead, a series of palmatine analogues with variations on the 9*O* atom (Figure 1) were achieved to elucidate the SAR and develop a novel class of anti-COL1A1 agents. Specifically, ester and ether derivatives of **2** were prepared and evaluated for their effect on inhibiting the activity of COL1A1 promotor, and the primary anti-COL1A1 mechanism exploration of the key compound was also carried out.

## 2. Results and Discussion

### 2.1. Chemistry

A total of 20 derivatives, among which compounds **4a–b**, **4f**, **4h–i**, **5**, **6b–e**, **6g**, and **6i–j** were firstly prepared from commercially available **2**, which was purchased from Xi’an Tianbao Biotechnology Co., Ltd. (Shanxi, China) with purity over 95%. Figure 2 depicts the synthetic routes.

As a start, **2** was heated at 195−210 °C under reduced pressure (30−40 mmHg) and acidified in concentrated HCl/ethanol (5/95 by vol.) to get the key intermediate **3** in an 80% yield while using the previous procedures [20]. Subsequently, esters **4a–j** and sulfonate **5** were acquired by esterification or sulfonation on the free hydroxyl motif of **3** in 43−68% yields [21]. Afterwards, the reaction of **3** and substituted benzyl halide or halogenated hydrocarbons under alkaline condition in the presence or absence of sodium iodide (NaI) achieved compounds **6a–j** in yields of 32–57% [22]. All of the targeted compounds were purified with flash column chromatography on silica gel while using CH_2_Cl_2_/MeOH as eluent.

### 2.2. Target Compounds Inhibited the Activity of COL1A1 Promotor in Human Hepatic Stellate LX-2 Cells

The COL1A1-based luciferase reporter model was applied to screen all of the target compounds (20 μM) in human hepatic stellate LX-2 cells, taking EGCG as the positive control [23]. The LX-2 cells were transfected with COL1A1 promotor (pGL4.17-COL1A1-Pro) by lipofactamine 2000 (Invitrogen) followed the standard protocol for 24 h, and then treated with the tested compound for another 24 h [11]. Table 1 displays the structures and inhibitory rates (%) of all target compounds.

Initially, *m*-fluoro and *p*-methoxy motifs were introduced on ring E of the lead **1,** respectively, to generate compounds **4a** and **4b**, and no obvious improvement on activity was achieved. Subsequently, the benzene E ring was replaced with a series of straight-chain alkane (**4c–e**) or a cycloalkane (**4f**), unfortunately, all of them showed no benefit on the anti-COL1A1 activity. Meanwhile, the replacement of benzene ring with aminos (**4g** and **4h**) also gave declined activity. Subsequently, the carbonyl moiety in compound **4g** was altered with its bioisosteric thiocarbonyl to generate compound **4i**, which gave an even lower activity. These results highlighted the importance of the benzene E ring.

Therefore, in the next round, ring E was retained, the impact of the ester linker was investigated. Firstly, the ester linker was replaced with sulfonate linker, and the corresponding compound **5** gave a comparable activity to the lead **1**. Afterwards, the ester linker was changed into ether, on which a series of benzyl (**6a–g**), alkyl (**6h** and **6i**) or allyl (**6j**) moiety was attached. Compound **6a** with a 9*O*-benzyloxy moiety displayed an elevated activity as compared with the lead **1**. Subsequently, electron-donating alkyls were attached on ring E, and the corresponding derivatives **6b–e** showed remarkably higher activities with the inhibition rates of 77.60%, 96.77%, 84.10%, and 96.53%, respectively. Therefore, the bulky alkyls might be more beneficial. As a comparison, the introduction of an electron-withdrawing nitro group (**6f** and **6g**) caused decreased activity when compared with **6a**. Meanwhile, the replacement of benzene E rings with an alkyl or allyl moiety (**6h–j**) only gave comparable or declined activity. These results indicated that the introduction of 9*O*-benzyloxy with a higher electron density on the benzene ring was beneficial for the activity.

The potency on COL1A1 promotor expression of the three most potent compounds **6c–e** was determined, and they gave half maximal inhibition concentration (IC_50_) values of 3.98 μM, 5.56 μM, and 4.47 μM, respectively. In the next cytotoxicity evaluation, compound **6c** displayed the highest cellular safety profile with the median cytotoxic concentration (CC_50_) value of 39.30 μM in LX-2 cell and, thus, gave the highest selectivity index (SI, CC_50_/IC_50_) value of 9.9, as indicated in Table 1.

### 2.3. Key Compounds Inhibited the Expression of COL1A1 in mRNA and Protein Levels

LX-2 cells were activated by TGF-β1 (2 ng/mL), and simultaneously treated with key compounds **6a–e** at the concentration of 10 μM, respectively, while taking **1** as the positive control. As expected, real-time PCR (RT-PCR) amplification results demonstrated that the stimulation of TGF-β1 caused a boomed transcription of *COL1A1*, which was significantly supressed by the administration of the tested compounds. The inhibition rates of compound **6a–e** were 77.09%, 37.11%, 93.24%, 73.78%, and 75.61%, respectively (Figure 3A). Among them, compound **6c** showed the highest potency.

Afterwards, their inhibitory effect on the protein expression of COL1A1 was investigated by western blot assay. As indicated in Figure 3B,C, all of these compounds (10 μM) significantly retarded the enrichment of COL1A1 protein induced by TGF-β1, and the boomed COL1A1 was virtually reversed by these compounds, including **6a–e**. These results suggested that these palmatine analogues could effectively reduce COL1A1 expression in both mRNA and protein levels.

### 2.4. Key Compounds Inhibited the Expressions of Fibrogenic Genes

The effects of **6a–e** on the expressions of a series of well-recognized fibrogenic genes were evaluated by the RT-PCR assay. As shown in Figure 4A, the elevated *ACTA2* (encode α-SMA) mRNA level was successfully achieved by TGF-β1 treatment, which was statistically repressed after the administration of indicated compounds. The inhibition rates were 52.47%, 38.79%, 63.05%, 37.07%, and 42.82%, respectively, being much higher than the lead **1**.

Matrix metalloprotein 2 (MMP2) is an endogenous peptidase for degrading the basement membrane, and its rapid development marks the formation of liver fibrosis [24]. Therein, the inhibitory effects on MMP2 of key compounds under TGF-β1 stimulation were respectively evaluated. As shown in Figure 4B, an approximately five-fold increase of *MMP2* mRNA was recorded upon the stimulation of TGF-β1, which could be retarded by the administration of lead **1**. To our great delight, the treatment of **6a–e** (10 μM) exhibited even stronger repressive effects to varied degrees, with inhibition rates of 36.52%, 43.00%, 86.43%, 65.39%, and 55.99%, respectively. Among them, compound **6c** exerted the highest potency, and almost abolished the TGF-β1-induced MMP2 expression in LX-2 cells.

TGF-β1, an important cytokine, plays a vital role in the progression of liver fibrogenesis. Its secretion stimulates the overexpression of fibrogenic genes, like COL1A1, α-SMA, MMP2, as well as TGF-β1 itself, also known as a cascade amplification effect [24,25]. Therefore, the blocking of TGF-β1 expression leads to the suppression of fibrogenesis. Based on this piece of knowledge, the inhibitory effects of **6a–e** on TGF-β1 were investigated. TGF-β1 treatment brought the booming expression of *TGFB1* mRNA with an anticipated success, which was effectively reversed by the indicated compounds, as shown in Figure 4C. RT-PCR analysis disclosed that **6a–e** gave the inhibitory rates of 49.35%, 21.19%, 74.38%, 50.20%, and 61.16%, respectively.

Next, the effects of **6a–e** on the expressions of these fibrogenic proteins were also evaluated by western blot assay. As shown in Figure 4D–G, the elevated expressions of α-SMA, MMP2, and TGF-β1 proteins under TGF-β1 stimulation were observed, as anticipated. These compounds showed extensive inhibitory effects on the indicated proteins to varying degrees and **6c** stood out in all assays, which is consistent with the above RT-PCR results.

As **6c** exhibited the most potent inhibitory effects against the expressions of all tested fibrogenic proteins, its dose-dependent inhibition effect and potential mechanism were carried out in the next step.

### 2.5. ***6c*** Inhibited the Expressions of Fibrogentic Proteins in a Dose-Dependent Manner

The dose-dependent anti-fibrogenic effect of **6c** was carried out by western blot assay, and a concentration gradient of 0 μM, 2.5 μM, 5 μM, and 10 μM was applied. **6c** significantly reversed the increase of COL1A1, α-SMA, MMP2, and TGF-β1 protein induced by TGF-β1 in a dose-dependent manner, and expressions of COL1A1 and TGF-β1 were nearly abolished at the concentration of 10 μM, as indicated in Figure 5A,C–F.

### 2.6. ***6c*** Supressed the JAK1/STAT3 Signaling Pathway

In the canonical Smad-dependent pathway, the activation of TGF-β1 leads to the phosphorylation of both Smad2 and Smad3, which was capable of translocating to the nucleus and regulating transcriptional responses. LX-2 cells were treated with TGF-β1 and a concentration gradient (0 μM, 2.5 μM, 5 μΜ, and 10 μM) of **6c** for 24 h. TGF-β1 treatment stimulated the phosphorylation of Smad2 and Smad3, which was down-regulated by **6c** at the concentration of 10 μM, demonstrating **6c**’s ability to repress the canonical TGF-β1/Smads pathway, as shown in Figure 5A. However, it caught our attention that the anti-COL1A1 IC_50_ value of **6c** was much lower than the effect-acting concentration, indicating the presence of other mechanism(s).

Recently, it has been discovered that some Smad-independent signaling pathways might regulate the activiation of HSCs by TGF-β1 [26]. For example, TGF-β1 was also demonstrated to stimulate the Janus kinase (JAK1)/signal transducer and the activator of transcription 3 (STAT3) signaling axis and induce liver fibrosis [27]. The expression level of STAT3, a crucial checkpoint for fibroblast activation and hepatic fibrosis [28], was tested to further understand the anti-fibrogenic mechanism of **6c**. Similarly, LX-2 cells were treated with TGF-β1 and the same concentration gradient of **6c,** as mentioned above. As indicated in Figure 5B,H–J, **6c** dose-dependently reversed the phosphorylation of STAT3 induced by TGF-β1 activation in a dose-dependent manner, without affecting the phosphorylation of STAT1 and total STAT3 itself. In addition, **6c** demonstrated a dose-dependent inhibitory effect on the phosphorylation of JAK1, being recognized as an upstream regulator for the phosphorylation of STAT3 [28]. Therefore, repressing the JAK1/STAT3 signaling pathway, might also contribute to **6c**’s anti-fibrogenic effect. However, repressing the JAK1/STAT3 signaling pathway might be the predominant mechanism since **6c** repressed the JAK1/STAT3 pathway at a lower concentration.

Taken together, it was speculated that **6c** might repress the liver fibrogenesis through initially blocking both canonical TGF-β1/Smads and non-canonical JAK1/STAT3 signaling pathways, then down-regulating the expressions of targeted fibrogenesis-associated genes, for example, COL1A1, α-SMA, and MMP2, as depicted in Figure 6.

### 2.7. Safety Profile of **6c**

An acute toxicity test was performed in Kunming mice to evaluate the safety profile of **6c**. **6c** was given orally in a single-dosing experiment at 0, 250, 500 or 1000 mg·kg^−1^, respectively. All of the mice survived during the seven-day observation period, with glossy hair, fleshy body, agile movement, and good appetite. Therefore, **6c** gave the median lethal dose (LD_50_) value over 1000 mg·kg^−1^, indicating a good safety profile in vivo.

### 2.8. Druglike Property Prediction of **6c**

Next, the druggability predition of **6c** was carried out, while using software ADMET Predictor version 9.5 (Simulations Plus Inc., Lancaster, CA, USA) [29]. The computer simulation results are listed in Table 2 and revealed that the absorption, metabolism, and toxicity properties of compound **6c** were all in reasonable range, which suggested that **6c** might own a reasonale druglike property.

## 3. Materials and Methods

### 3.1. Apparatus, Materials, and Analysis Reagents

All of the chemical reagents and anhydrous solvents were obtained from commercial sources and used without further purification. MPA100 OptiMelt automated melting point system (Stanford Research Systems, Palo Alto, CA, USA) were used to acquire the Melting points (m.p.). ^1^H and ^13^C-NMR spectra were recorded on a Bruker Avance 600 MHz spectrometer (AV600-III, Bruker, Zürich, Swiss) in DMSO-*d_6_* with TMS as the internal standard. ESI and ESI high-resolution mass spectrometry (HRMS) was performed on an Autospec Uitima-TOF mass spectrometer (Micromass UK Ltd., Manchester, UK). Flash chromatography was performed on Combiflash Rf 200 (Teledyne, Lincoln, NE, USA) with an average particle size of 0.038 mm.

The syntheis of target compounds **4c–e**, **4g**, **6a**, **6f**, and **6h** were reported previously [30,31].

### 3.2. Chemistry

#### 3.2.1. General Procedure for the Synthesis of Compounds **4a–i** and **5**

**2** (3.87 g, 10 mmol) was heated at 195–210 °C for 30 min. under vacuum (20–30 mmHg) to afford the deep purple solid, which was acidified with ethanol/concentrated HCl (95/5) and the yellow solution was acquired. The solvent was removed, and the residue was purified by flash chromatography while using CH_2_Cl_2_/CH_3_OH as the gradient eluent, affording the intermediate **3** (3.43 g, 80%) as a brown solid.

Triethylamine (1.80 mmol) was added to a stirred solution of **3** (100 mg, 0.40 mmol) in anhydrous CH_3_CN (6 mL) first, 10 min. later, corresponding acyl chloride (1.2 mmol) or sulfonyl chloride (1.2 mmol) were added in the reaction system. The reaction mixture was heated at 65 °C for 3–8 h. The system was cooled, filtered, and the resulting residue was purified by flash chromatography using CH_2_Cl_2_/CH_3_OH as the gradient eluent, to acquire desired compounds.

*2,3,10-Trimethoxy-9-**m-fluorophenylacetoxylprotopalmatine chloride* (**4a**): Total yield: 61%; yellow solid; m.p.: 204–206 °C. ^1^H-NMR: *δ* 10.05 (d, *J* = 4.8 Hz, 1H), 9.14 (s, 1H), 8.28 (d, *J* = 9.0 Hz, 1H), 8.24 (d, *J* = 9.0 Hz, 1H), 7.74 (s, 1H), 7.46 (q, *J* = 7.8 Hz, 1H), 7.39–7.29 (m, 2H), 7.18 (t, *J* = 9.0 Hz, 1H), 7.11 (s, 1H), 4.99 (t, *J* = 6.6 Hz, 2H), 4.35 (s, 2H), 4.00 (s, 3H), 3.95 (s, 3H), 3.88 (s, 3H), 3.26 (t, *J* = 6.6 Hz, 2H); ^13^C-NMR: *δ* 168.6, 162.1, 151.7, 150.2, 148.8, 144.5, 138.4, 136.2, 133.3, 133.0, 130.3 128.8, 126.8, 125.9 (2), 121.0, 120.3, 118.8, 116.6, 116.5, 114.1, 111.3, 108.9, 57.2, 56.2, 55.9, 55.5, 25.8; HRMS: calcd for C_28_H_25_NO_5_FCl [M − Cl]^+^: 474.1711, found: 474.1710.

*2,3,10-Trimethoxy-9-p-methoxyphenylacetoxylprotopalmatine chloride* (**4b**): Total yield: 58%; yellow solid; m.p.: 208–210 °C. ^1^H-NMR: *δ* 10.03 (s, 1H), 9.16 (s, 1H), 8.27 (d, *J* = 9.6 Hz, 1H), 8.23 (d, *J* = 9.0 Hz, 1H), 7.74 (s, 1H), 7.42–7.31 (m, 2H), 7.11 (s, 1H), 7.03–6.86 (m, 2H), 5.00 (t, *J* = 6.6 Hz, 2H), 4.22 (s, 2H), 3.99 (s, 3H), 3.95 (s, 3H), 3.88 (s, 3H), 3.77 (s, 3H), 3.26 (t, *J* = 6.6 Hz, 2H); ^13^C-NMR: *δ* 169.3, 158.4, 151.6, 150.2, 148.7, 144.4, 138.3, 133.5, 133.0, 130.8 (2), 128.8, 126.6, 125.9, 125.4, 121.0, 120.3, 118.8, 113.8 (2), 111.3, 108.9, 57.2, 56.2, 55.9, 55.5, 55.1, 38.9, 25.8; HRMS: calcd for C_29_H_28_NO_6_Cl [M − Cl]^+^: 486.1911, found: 4486.1911.

*2,3,10-Trimethoxy-9-**n-propionyloxyprotopalmatine chloride* (**4c**): Total yield: 68%; yellowish-brown solid; m.p.: 186–188 °C. ^1^H-NMR: *δ* 9.94 (s, 1H), 9.15 (s, 1H), 8.31–8.27 (m, 1H), 8.24 (d, *J* = 9.6 Hz, 1H), 7.74 (s, 1H), 7.10 (s, 1H), 4.97 (t, *J* = 6.6 Hz, 2H), 4.04 (s, 3H), 3.95 (s, 3H), 3.87 (s, 3H), 3.24 (t, *J* = 6.6 Hz, 2H), 2.91 (q, *J* = 7.8 Hz, 2H), 1.24 (t, *J* = 7.8 Hz, 3H); ^13^C-NMR: *δ* 171.5, 151.6, 150.3, 148.7, 144.4, 138.3, 133.5, 133.0, 128.8, 126.5, 125.9, 121.1, 120.3, 118.8, 111.3, 108.9, 57.2, 56.2, 55.9, 55.5, 26.6, 25.8, 8.8; HRMS: calcd for C_23_H_24_NO_5_Cl [M − Cl]^+^: 394.1649, found: 394.1649.

*2,3,10-Trimethoxy-9-**n-butyryloxyprotopalmatine chloride* (**4d**): Total yield: 57%; yellow solid; m.p.: 197–199 °C. ^1^H-NMR: *δ* 9.94 (s, 1H), 9.17 (s, 1H), 8.29 (d, *J* = 9.0 Hz, 1H), 8.24 (d, *J* = 9.0 Hz, 1H), 7.75 (s, 1H), 7.10 (s, 1H), 4.98 (t, *J* = 6.6 Hz, 2H), 4.03 (s, 3H), 3.95 (s, 3H), 3.88 (s, 3H), 3.24 (t, *J* = 6.6 Hz, 2H), 2.86 (t, *J* = 7.2 Hz, 2H), 1.77 (q, *J* = 7.2 Hz, 2H), 1.06 (t, *J* = 7.2 Hz, 3H); ^13^C-NMR: *δ* 170.5, 151.6, 150.2, 148.7, 144.4, 138.3, 133.5, 133.0, 128.8, 126.5, 125.9, 121.1, 120.3, 118.8, 111.3, 108.9, 57.2, 56.2, 55.9, 55.5, 35.0, 25.8, 17.8, 13.4; HRMS: calcd for C_24_H_26_NO_5_Cl [M − Cl]^+^: 408.1806, found: 408.1804.

*2,3,10-Trimethoxy-9-**n-valeryloxyprotopalmatine chloride* (**4e**): Total yield: 62%; yellow solid; m.p.: 206–208 °C. ^1^H-NMR: *δ* 9.95 (s, 1H), 9.17 (s, 1H), 8.29 (d, *J* = 9.0 Hz, 1H), 8.24 (d, *J* = 9.0 Hz, 1H), 7.75 (s, 1H), 7.10 (s, 1H), 4.98 (t, *J* = 6.6 Hz, 2H), 4.03 (s, 3H), 3.95 (s, 3H), 3.87 (s, 3H), 3.24 (t, *J* = 6.6 Hz, 2H), 2.88 (t, *J* = 7.2 Hz, 2H), 1.80–1.65 (m, 2H), 1.57–1.36 (m, 2H), 0.98 (t, *J* = 7.2 Hz, 3H); ^13^C-NMR: *δ* 170.7, 151.6, 150.2, 148.7, 144.4, 138.3, 133.5, 133.0, 128.8, 126.5, 125.9, 121.1, 120.3, 118.8, 111.3, 108.9, 57.2, 56.2, 55.9, 55.5, 32.9, 26.3, 25.8, 21.5, 13.7; HRMS: calcd for C_25_H_28_NO_5_Cl [M − Cl]^+^: 422.1962, found: 422.1961.

*2,3,10-Trimethoxy-9-cyclopentylacetoxylprotopalmatine chloride* (**4f**): Total yield: 49%; yellow solid; m.p.: 208–210 °C. ^1^H-NMR: *δ* 9.92 (s, 1H), 9.16 (s, 1H), 8.29 (d, *J* = 9.0 Hz, 1H), 8.24 (d, *J* = 9.0 Hz, 1H), 7.74 (s, 1H), 7.10 (s, 1H), 4.97 (t, *J* = 6.6 Hz, 2H), 4.03 (s, 3H), 3.95 (s, 3H), 3.88 (s, 3H), 3.25 (t, *J* = 6.6 Hz, 2H), 2.88 (d, *J* = 7.8 Hz, 2H), 2.37 (p, *J* = 7.8 Hz, 1H), 1.96–1.87 (m, 2H), 1.72–1.65 (m, 2H), 1.61–1.55 (m, 2H), 1.38–1.30 (m, 2H); ^13^C-NMR: *δ* 170.0, 151.6, 150.2, 148.7, 144.4, 138.3, 133.6, 133.0, 128.8, 126.5, 125.9, 121.1, 120.3, 118.8, 111.3, 108.9, 57.2, 56.2, 55.9, 55.5, 35.9, 31.9 (2), 25.8, 24.7 (2); HRMS: calcd for C_27_H_30_NO_5_Cl [M − Cl]^+^: 448.2119, found: 448.2118.

*2,3,10-Trimethoxy-9-N,N-dimethylaminocarbonyloxyprotopalmatine chloride* (**4g**): Total yield: 47%; brown solid; m.p.: 195–197 °C. ^1^H-NMR: *δ* 9.89 (s, 1H), 9.16 (s, 1H), 8.27 (d, *J* = 9.0 Hz, 1H), 8.21 (d, *J* = 9.0 Hz, 1H), 7.75 (s, 1H), 7.10 (s, 1H), 5.00 (t, *J* = 6.6 Hz, 2H), 4.03 (s, 3H), 3.95 (s, 3H), 3.87 (s, 3H), 3.28–3.21 (m, 5H), 3.01 (s, 3H); ^13^C-NMR: *δ* 153.1, 152.0, 151.2, 149.2, 144.9, 138.5, 135.0, 133.4, 129.2, 126.5, 126.4, 122.1, 120.7, 119.3, 111.7, 109.3, 57.6, 56.6, 56.3, 55.9, 37.2, 37.0, 26.3; HRMS: calcd for C_23_H_25_N_2_O_5_Cl [M − Cl]^+^: 409.1758, found: 409.1758.

*2,3,10-Trimethoxy-9-(3′-(methylsulfonyl)-2′-oxoimidazolidine-1′-carbonyl)oxyprotopalmatine chloride* (**4h**): Total yield: 43%; yellow solid; m.p.: 180–182 °C. ^1^H-NMR: *δ* 9.97 (s, 1H), 9.15 (s, 1H), 8.34 (d, *J* = 9.0 Hz, 1H), 8.28 (d, *J* = 9.0 Hz, 1H), 7.74 (s, 1H), 7.11 (s, 1H), 4.95 (t, *J* = 6.6 Hz, 2H), 4.16 (s, 2H), 4.08 (s, 3H), 3.99 (t, *J* = 7.8 Hz, 2H), 3.95 (s, 3H), 3.88 (s, 3H), 3.44 (s, 3H), 3.26 (t, *J* = 6.6 Hz, 2H); ^13^C-NMR: *δ* 151.7, 150.5, 149.6, 148.8, 144.3, 138.6, 133.1, 132.2, 129.6, 128.8, 127.2, 126.0, 121.0, 120.3, 118.8, 111.3, 108.9, 57.3, 56.2, 55.9, 55.7, 41.1, 40.6, 31.3, 25.8; HRMS: calcd for C_25_H_26_N_3_O_8_SCl [M − Cl]^+^: 528.1435, found: 528.1435.

*2,3,10-Trimethoxy-9-N,N-dimethylcarbamothioyloxyprotopalmatine chloride* (**4i**): Total yield: 52%; brown solid; m.p.: 233–235 °C. ^1^H-NMR: *δ* 9.65 (s, 1H), 8.96 (s, 1H), 8.18 (s, 2H), 7.66 (s, 1H), 7.03 (s, 1H), 4.93 (t, J = 6.6 Hz, 2H), 4.00 (s, 3H), 3.92 (s, 3H), 3.86 (s, 3H), 3.47 (s, 3H), 3.43 (s, 3H), 3.30–3.16 (m, 2H); ^13^C-NMR: *δ* 186.2, 152.7, 151.8, 149.7, 145.1, 139.1, 137.5, 133.8, 129.4, 127.1, 126.6, 122.9, 121.0, 119.5, 111.7, 109.4, 57.5, 56.5, 56.3, 56.2, 43.7 (2), 26.7; HRMS: calcd for C_23_H_25_N2O_4_SCl [M − Cl]^+^: 425.1530, found: 425.1529.

*2,3,10-Trimethoxy-9-p-nitrophenylsulfonyloxyprotopalmatine p-nitro benzenesulfonate* (**5**): Total yield: 49%; yellowish-brown solid; m.p.: 239–241 °C. ^1^H-NMR: *δ* 9.66 (s, 1H), 9.16 (s, 1H), 8.56–8.48 (m, 2H), 8.30 (d, *J* = 9.0 Hz, 1H), 8.29–8.25 (m, 2H), 8.22 (d, *J* = 9.0 Hz, 1H), 8.20–8.17 (m, 2H), 7.84–7.80 (m, 2H), 7.73 (s, 1H), 7.13 (s, 1H), 4.98 (t, *J* = 6.6 Hz, 2H), 3.94 (s, 3H), 3.89 (s, 3H), 3.70 (s, 3H), 3.24 (t, *J* = 6.6 Hz, 2H); ^13^C-NMR: *δ* 154.3, 151.9, 151.3, 151.2, 148.8, 147.2, 143.7, 140.0, 139.1, 133.5, 130.7, 130.3 (2), 129.0, 128.7, 126.9 (2), 126.1, 124.9 (2), 123.3 (2), 121.5, 120.7, 118.6, 111.3, 108.9, 56.8, 56.2, 55.9, 55.9, 25.8; HRMS: calcd for C_32_H_27_N_3_O_12_S_2_ [M – C_2_H_4_NO_4_S]^+^: 523.1170, found: 523.1170.

#### 3.2.2. General Procedure for the Synthesis of **6a–j**

To a stirred solution of compound **3** (100 mg, 0.40 mmol) in anhydrous DMF (6 mL) were added K_2_CO_3_ (1.60 mmol) or K_2_CO_3_ (1.60 mmol) and NaI (1.60 mmol), 10 min. later, benzyl halide (1.6 mmol) or halogenated hydrocarbon (1.6 mmol) were added in the reaction system. The reaction mixture was heated at 71 °C for 2–24 h. The reaction system was cooled, filtered and the resulting residue was purified by flash chromatography over silica gel using CH_2_Cl_2_/CH_3_OH as the gradient eluent, to acquired desired compounds.

*2,3,10-Trimethoxy-9-benzyloxyprotopalmatine**bromide* (**6a**) [20]: Total yield: 52%; yellow solid; m.p.: 223–225 °C. ^1^H-NMR: *δ* 9.73 (s, 1H), 9.03 (s, 1H), 8.23 (dd, *J* = 9.0, 1.8 Hz, 1H), 8.04 (d, *J* = 9.0 Hz, 1H), 7.71 (s, 1H), 7.59 (d, *J* = 7.8 Hz, 2H), 7.44–7.31 (m, 3H), 7.10 (s, 1H), 5.36 (s, 2H), 4.93 (t, *J* = 6.6 Hz, 2H), 4.09 (s, 3H), 3.93 (s, 3H), 3.87 (s, 3H), 3.21 (t, *J* = 6.6 Hz, 2H); ^13^C-NMR: *δ* 151.9, 151.0, 149.2, 145.8, 142.4, 138.0, 136.9, 133.5, 129.2 (2), 129.0, 128.9, 128.8 (2), 127.0, 124.1, 122.2, 120.3, 119.3, 111.7, 109.2, 75.8, 57.5, 56.6, 56.3, 56.0, 26.4; ESI-MS: [M − Br]^+^: 428.

*2,3,10-Trimethoxy-9-p-methylbenzyloxyprotopalmatine**bromide* (**6b**): Total yield: 57%; yellow solid; m.p.: 226–228 °C. ^1^H-NMR: *δ* 9.70 (s, 1H), 9.04 (s, 1H), 8.22 (d, *J* = 9.0 Hz, 1H), 8.03 (d, *J* = 9.0 Hz, 1H), 7.71 (s, 1H), 7.47 (d, *J* = 8.4 Hz, 2H), 7.20 (d, *J* = 7.8 Hz, 2H), 7.10 (s, 1H), 5.32 (s, 2H), 4.92 (t, *J* = 6.6 Hz, 2H), 4.09 (s, 3H), 3.94 (s, 3H), 3.87 (s, 3H), 3.21 (t, *J* = 6.6 Hz, 2H), 2.29 (s, 3H); ^13^C-NMR: *δ* 151.5, 150.6, 148.7, 145.3, 141.9, 137.8, 137.6, 133.4, 133.0, 129.0 (2), 128.9 (2), 128.6, 126.7, 123.5, 121.8, 119.9, 118.9, 111.3, 108.7, 75.2, 57.0, 56.2, 55.9, 55.5, 26.0, 20.8; HRMS: calcd for C_28_H_28_NO_4_Br [M − Br]^+^: 442.2013, found: 442.2012.

*2,3,10-Trimethoxy-9-p-isopropyloxyprotopalmatine**bromide* (**6c**): Total yield: 49%; yellow solid; m.p.: 218–220 °C. ^1^H-NMR: *δ* 9.70 (s, 1H), 9.05 (s, 1H), 8.23 (d, *J* = 9.0 Hz, 1H), 8.05 (d, *J* = 9.0 Hz, 1H), 7.71 (s, 1H), 7.63–7.45 (m, 2H), 7.37–7.23 (m, 2H), 7.10 (s, 1H), 5.32 (s, 2H), 4.93 (t, *J* = 6.6 Hz, 2H), 4.09 (s, 3H), 3.94 (s, 3H), 3.88 (s, 3H), 3.22 (t, *J* = 6.6 Hz, 2H), 2.89 (p, *J* = 6.6 Hz, 1H), 1.19 (d, *J* = 6.6 Hz, 6H); ^13^C-NMR: *δ* 152.0, 151.1, 149.2 (2), 145.8, 142.6, 138.1, 134.4, 133.5, 129.4 (2), 129.1, 127.1, 126.7 (2), 124.1, 122.3, 120.4, 119.4, 111.8, 109.3, 75.8, 57.5, 56.7, 56.4, 56.0, 33.7, 26.5, 24.3 (2); HRMS: calcd for C_30_H_32_NO_4_Br [M − Br]^+^: 470.2326, found: 470.2326.

*2,3,10-Trimethoxy-9-p-tert-butylbenzyloxyprotopalmatine**bromide* (**6d**): Total yield: 36%; yellow solid; m.p.: 198–200 °C. ^1^H-NMR: *δ* 9.71 (s, 1H), 9.06 (s, 1H), 8.23 (d, *J* = 9.6 Hz, 1H), 8.06 (d, *J* = 9.0 Hz, 1H), 7.72 (s, 1H), 7.61–7.48 (m, 2H), 7.48–7.34 (m, 2H), 7.10 (s, 1H), 5.32 (s, 2H), 4.93 (t, *J* = 6.6 Hz, 2H), 4.09 (s, 3H), 3.94 (s, 3H), 3.88 (s, 3H), 3.22 (t, *J* = 6.6 Hz, 2H), 1.28 (s, 9H); ^13^C-NMR: *δ* 152.0, 151.4, 151.1, 149.2, 145.8, 142.6, 138.1, 134.0, 133.6, 129.1 (2), 129.0, 127.1, 125.5 (2), 124.1, 122.3, 120.4, 119.4, 111.8, 109.3, 75.7, 57.5, 56.7, 56.4, 56.0, 34.8, 31.6 (3), 26.5; HRMS: calcd for C_31_H_34_NO_4_Br [M − Br]^+^: 484.2482, found: 484.2482.

*2,3,10-Trimethoxy-9-3′,5′-di-tert-butylbenzyloxyprotopalmatine**bromide* (**6e**): Total yield: 32%; yellow solid; m.p.: 187–189 °C. ^1^H-NMR: *δ* 9.65 (s, 1H), 9.02 (s, 1H), 8.22 (d, *J* = 9.0 Hz, 1H), 8.02 (d, *J* = 9.0 Hz, 1H), 7.69 (s, 1H), 7.31 (t, *J* = 1.8 Hz, 1H), 7.29–7.28 (m, 2H), 7.09 (s, 1H), 5.37 (s, 2H), 4.84 (t, *J* = 6.6 Hz, 2H), 4.10 (s, 3H), 3.93 (s, 3H), 3.87 (s, 3H), 3.17 (t, *J* = 6.6 Hz, 2H), 1.21 (s, 18H); ^13^C-NMR: *δ* 151.5, 150.8, 150.3 (2), 148.8, 145.5, 141.8, 137.4, 135.1, 132.9, 128.4, 126.4, 123.6, 123.4 (2), 122.3, 121.9, 119.9, 118.8, 111.3, 108.7, 76.1, 57.0, 56.2, 55.9, 55.5, 34.4 (2), 31.2 (6), 26.0; HRMS: calcd for C_35_H_42_NO_4_Br [M − Br]^+^: 540.3108, found: 540.3109.

*2,3,10-Trimethoxy-9-p-nitrobenzyloxyprotopalmatine**bromide* (**6f**): Total yield: 47%; brown solid; m.p.: 176–178 °C. ^1^H-NMR: *δ* 9.86 (s, 1H), 9.07 (s, 1H), 8.30 (dd, *J* = 8.4, 2.4 Hz, 2H), 8.24 (dd, *J* = 9.0, 1.9 Hz, 1H), 8.08 (d, *J* = 9.0 Hz, 1H), 7.90 (d, *J* = 8.4 Hz, 2H), 7.73 (s, 1H), 7.11 (d, *J* = 1.8 Hz, 1H), 5.51 (s, 2H), 4.96 (t, *J* = 6.6 Hz, 2H), 4.07 (s, 3H), 3.94 (s, 3H), 3.88 (s, 3H), 3.23 (t, *J* = 6.6 Hz, 2H); ^13^C-NMR: *δ* 151.5, 150.3, 148.7, 147.2, 145.3, 144.4, 141.7, 137.8, 133.1, 129.1 (2), 128.7, 126.6, 123.9, 123.5 (2), 121.5, 119.9, 118.9, 111.3, 108.8, 74.0, 57.1, 56.2, 55.9, 55.5, 26.0; HRMS: calcd for C_27_H_25_N_2_O_6_Br [M − Br]^+^: 473.1707, found: 473.1707.

*2,3,10-Trimethoxy-9-o-nitrobenzyloxyprotopalmatine**bromide* (**6g**): Total yield: 50%; yellow solid; m.p.: 234–236 °C. ^1^H-NMR *δ* 9.79 (s, 1H), 9.09 (s, 1H), 8.24 (d, *J* = 9.0 Hz, 1H), 8.19 (dd, *J* = 8.4, 1.2 Hz, 1H), 8.17–8.12 (m, 1H), 8.10 (d, *J* = 9.0 Hz, 1H), 7.91 (td, *J* = 7.8, 1.2 Hz, 1H), 7.77–7.58 (m, 2H), 7.10 (s, 1H), 5.68 (s, 2H), 4.91 (t, *J* = 6.6 Hz, 2H), 4.00 (s, 3H), 3.95 (s, 3H), 3.87 (s, 3H), 3.22 (t, *J* = 6.6 Hz, 2H); ^13^C-NMR: *δ* 151.5, 150.4, 148.7, 147.1, 145.3, 141.6, 137.8, 134.3, 133.2, 132.6, 129.5, 129.3, 128.6, 126.6, 124.7, 124.2, 121.4, 120.0, 118.9, 111.3, 108.8, 71.8, 57.0, 56.2, 55.9, 55.6, 26.0; HRMS: calcd for C_27_H_25_N_2_O_6_Br [M − Br]^+^: 473.1707, found: 473.1707.

*2,3,10-Trimethoxy-9-ethoxyprotopalmatine**bromide* (**6h**): Total yield: 43%; yellow solid; m.p.: 233–235 °C. ^1^H-NMR: *δ* 9.78 (s, 1H), 9.00 (s, 1H), 8.18 (d, *J* = 9.0 Hz, 1H), 8.00 (d, *J* = 9.0 Hz, 1H), 7.68 (s, 1H), 7.07 (s, 1H), 4.94 (t, *J* = 6.6 Hz, 2H), 4.34 (q, *J* = 7.2 Hz, 2H), 4.03 (s, 3H), 3.91 (s, 3H), 3.85 (s, 3H), 3.21 (t, *J* = 6.6 Hz, 2H), 1.43 (t, *J* = 7.2 Hz, 3H); ^13^C-NMR: *δ* 151.5, 150.4, 148.7, 145.4, 142.5, 137.7, 133.1, 128.6, 126.7, 123.2, 121.8, 119.8, 118.9, 111.3, 108.7, 69.9, 57.0, 56.2, 55.9, 55.4, 26.0, 15.4; HRMS: calcd for C_22_H_24_NO_4_Br [M − Br]^+^: 366.1700, found: 366.1698.

*2,3,10-Trimethoxy-9-propoxyprotopalmatine**bromide* (**6i**): Total yield: 38%; yellow solid; m.p.: 201–203 °C. ^1^H-NMR: *δ* 9.74 (s, 1H), 9.03 (s, 1H), 8.21 (dd, *J* = 9.6, 2.4 Hz, 1H), 8.03 (d, *J* = 9.0 Hz, 1H), 7.72 (d, *J* = 2.4 Hz, 1H), 7.10 (s, 1H), 4.97 (t, *J* = 6.6 Hz, 2H), 4.30 (t, *J* = 6.6 Hz, 2H), 4.06 (s, 3H), 3.94 (s, 3H), 3.88 (s, 3H), 3.23 (t, *J* = 6.6 Hz, 2H), 1.92–1.82 (m, 2H), 1.53 (qd, *J* = 7.8, 2.4 Hz, 2H), 0.99 (td, *J* = 7.8, 2.4 Hz, 3H); ^13^C-NMR: *δ* 151.5, 150.2, 148.7, 145.3, 142.8, 137.7, 133.1, 128.6, 126.7, 123.1, 121.6, 119.9, 118.9, 111.3, 108.7, 73.9, 57.0, 56.2, 55.9, 55.5, 31.6, 26.0, 18.6, 13.8; HRMS: calcd for C_24_H_28_NO_4_Br [M − Br]^+^: 394.2013, found: 394.2014.

*2,3,10-Trimethoxy-9-allyloxyprotopalmatine**bromide* (**6j**): Total yield: 44%; yellowish-brown solid; m.p.: 136–138 °C. ^1^H-NMR: *δ* 9.82 (s, 1H), 9.06 (s, 1H), 8.25–8.18 (m, 1H), 8.04 (d, *J* = 9.0 Hz, 1H), 7.72 (s, 1H), 7.10 (s, 1H), 6.27–6.20 (m, 1H), 5.44 (dq, *J* = 17.4, 1.8 Hz, 1H), 5.29 (dd, *J* = 10.2, 1.8 Hz, 1H), 4.97 (t, *J* = 6.6 Hz, 2H), 4.92 – 4.75 (m, 2H), 4.07 (d, *J* = 3.0 Hz, 3H), 3.94 (s, 3H), 3.87 (s, 3H), 3.23 (t, *J* = 6.6 Hz, 2H); ^13^C-NMR: *δ* 151.5, 150.3, 148.7, 145.4, 142.0, 137.7, 133.8, 133.1, 128.6, 126.6, 123.4, 121.8, 119.9, 119.1, 118.9, 111.3, 108.8, 74.5, 57.0, 56.2, 55.9, 55.5, 26.0; HRMS: calcd for C_23_H_24_NO_4_Br [M − Br]^+^: 378.1700, found: 378.1701.

### 3.3. Biology Assay

#### 3.3.1. Cell Culture and Screening of Compounds

The human hepatic stellate LX-2 cells were cultured in DMEM/GlutaMAX I medium (Invitrogen, Carlsbad, CA, USA) with 10% FBS and 1% penicillin/streptomycin and incubated in a humidified atmosphere with 5% CO_2_ at 37 °C. 4 × 10^5^ cells were seeded in six-well plates and then transfected with pGL4.17-COL1A1-Pro plasmid for 24 h. Subsequently, the LX-2 cells were digested and re-seeded in 96-well plates at the density of 2 × 10^4^/well, and followed by palmatine derivatives (20 μM or 10 μM) treatment for another 24 h. At last, the COL1A1 promotor activity was determined while using the Bright-Glo luciferase assay system (Promega, Madison, WI, USA).

#### 3.3.2. Cell Survival Assay

The human hepatic stellate LX-2 cells were cultured in DMEM/GlutaMAX I medium with 10% FBS and 1% penicillin/streptomycin and then incubated in a humidified atmosphere with 5% CO_2_ at 37 °C. The cells were seeded in 96-well plates at the density of 2 × 10^4^/well and treated by the indicated palmatine derivatives for 24 h. After treatment, the cells were washed with PBS for three times and then fixed with 10% (*wt*/*v*) trichloroacetic acid for 1 h. Afterwards, the cells were stained with SRB for 30 min. After the excess SRB dye is removed by washing repeatedly with 1% (*v*/*v*) acetic acid, the protein-bound dye is dissolved in 10 mM Tris base solution for OD determination at 510 nm whlie using a microplate reader.

#### 3.3.3. RT-PCR Assay

The human hepatic stellate LX-2 cells were cultured in DMEM/GlutaMAX I medium (Invitrogen, USA) with 10% FBS and 1% penicillin/streptomycin and incubated in a humidified atmosphere with 5% CO_2_ at 37 °C. 4 × 10^5^ cells were seeded in six-well plates. When at 90–95% confluence, the cells were starvation with serum-free culture for 24 h. Subsequently, the cells were treated with TGF-β1 (2 ng/mL) and palmatine derivatives (10 μM) for another 24 h. Total RNA from the LX-2 cells was extracted while using Trizol reagent and purified by NucleoSpin RNA Clean-up. Reverse transcription was performed with Transcriptor first strand cDNA synthesis kit. The cDNA was then analyzed by ABI 7500 Fast Real-Time PCR System while using TaqMan probes of *TGF**B1*, *COL1A1*, *ACTA2*, *MMP2*, and *GAPDH* (ABI) and FastStart Universal Probe master mix (Roche, Indianapolis, IN, USA).

#### 3.3.4. Western blot

The LX-2 cells were cultured and treated, as described above. Briefly, the cells were washed with PBS and lysed in RIPA lysis for 30 min. in 4 °C; the supernatant was collected after centrifugation at 12,000 *g*, 4 °C for 15 min. Equal amounts of protein were quantified with Bradford assay, separated by SDS-PAGE, and transferred to polyvinylidene difluoride membrane. The membranes were blocked for 1 h at room temperature in PBST containing 5% milk and probed with specific first antibodies overnight at 4 °C. The membrane was washed three times by PBST, followed by horseradish peroxidase-conjugated secondary antibodies. The proteins were visualized using chemiluminescence reagents. The antibodies used in western blot analysis were obtained from Abcam(collagen 1 antibody (ab34710), a-SMA antibody (ab32575), TGF-β1 antibody (ab179695); Cell Signaling Technology (Phospho-STAT3(Tyr705) antibody (9131), STAT3 antibody(79D7), Phospho-STAT1 (Tyr701) antibody (58D6), Phospho-JAK1(Tyr1034/1035) antibody (74129), JAK1 antibody (3332) Phospho-Smad2/3 antibody (8828), Smad2 antibody (3122), GAPDH antibody (5174)); and, Proteintech (MMP2 antibody (10373-2-AP)).

#### 3.3.5. Acute Toxicity

Twenty-four female Kunming mice with weights of 20.0 ± 1.0 g were obtained from the Institute of Laboratory Animal Science (Beijing, China). The animals were kept according to the institutional guidelines of the Institute of Materia Medica, CAMS&PUMC. The mice were fed with regular rodent chow and housed in an air-conditioned room. The mice were randomly divided into different groups with six mice each. The compound was given orally in a single-dosing experiment at 0, 250, 500, or 1000 mg·kg^−1^ (saline as control), respectively. The mice were closely monitored for seven days. Body weight as well as survival was monitored.

#### 3.3.6. Statistics

The results are presented as mean values ± standard error of independent triplicate experiments. Statistical significant differences between groups were analyzed while using one-way analysis of variance (ANOVA), followed by Student’s t-test, and p-values of less than 0.05 were considered to be statistically significant.

## 4. Conclusions

To sum up, a series of 9*O*-substituted palmatine derivatives were designed, synthesized, and evaluated for their inhibitory effect on the COL1A1 promotor. The SAR indicated that the introduction of 9*O*-benzyl motif was beneficial for activity, and the electron density on the benzene ring might affect the potency. Among them, compound **6c** gave the most promising inhibitory effect against COL1A1, with an IC_50_ value of 3.98 μM. Its inhibition activity against COL1A1 was further confirmed on both mRNA and protein levels. Additionally, it dose-dependently inhibited the expressions of a series of fibrogenic genes and proteins, such as α-SMA and MMP2, indicating a promise against liver fibrogenesis. Further study indicated that it might exert the anti-fibrogenic effect via repressing both canonical TGF-β1/Smads and non-canonical JAK1/STAT3 signaling pathways. In addition, **6c** owned a high safety profile, with the LD_50_ value of over 1000 mg·kg^−1^ in mice as well as reasonable druglike property. Overall, this study explored the application of palmatine derivatives as a novel class of anti-fibrogenic agents for the very first time and therein offered powerful information for further structure optimization, and compound **6c** has been chosen as an ideal anti-hepatic fibrosis lead.

## Figures and Tables

**Figure 1 molecules-25-00773-f001:**
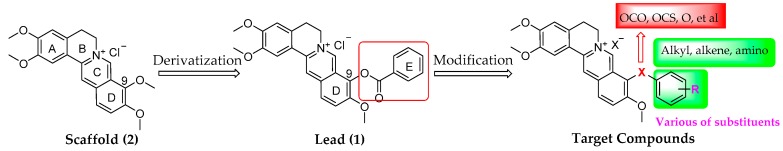
Chemical structures of **1**, palmatine Scaffold **2** and structural modification strategy.

**Figure 2 molecules-25-00773-f002:**
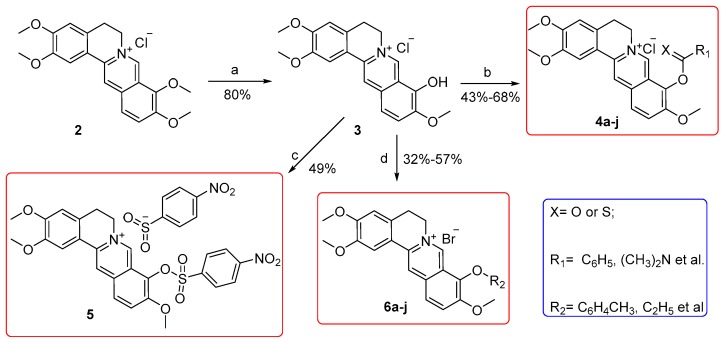
Reagents and conditions: (**a**) 195−210 °C, 30−40 mmHg, 30 min.; (**b**) RCOCl/RCSCl, triethylamine, CH_3_CN, 65 °C, 3−8 h; (**c**) 4-Nitrobenzenesulfonyl chloride, triethylamine, CH_3_CN, 65 °C, 4 h; (**d**) Substituted benzyl halide or halogenated hydrocarbons, K_2_CO_3_ for **6a–g** and **6j** or K_2_CO_3_ and NaI for **6h–i**, CH_3_CN, 71 °C, 2−24 h.

**Figure 3 molecules-25-00773-f003:**
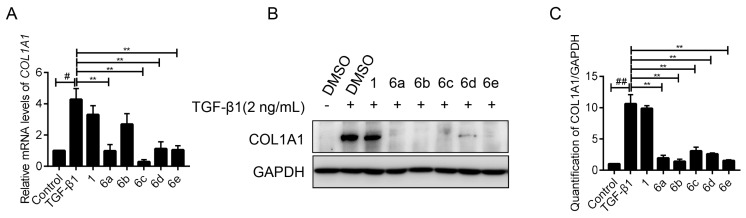
Inhibition effect of compounds on COL1A1 expression. The mRNA level (**A**) and protein level (**B**) of COL1A1 were detected in LX-2 cells after treatment with the indicated compound (10 μM), together with transforming growth factor-beta 1 (TGF-β1) (2 ng/mL) for 24 h. (**C**) Quantification of COL1A1 intensity of (**B**). The mRNA/protein level was normalized against GAPDH. Data were presented as the mean ± SEM, ^#^
*p* < 0.05, ^##^
*p* < 0.01 as compared to that of control group; ** *p* < 0.01 as compared to that of TGF-β1 group.

**Figure 4 molecules-25-00773-f004:**
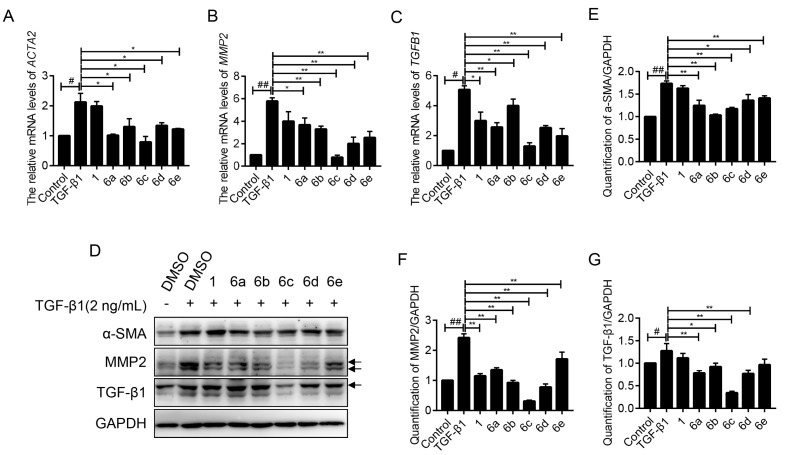
Inhibition effects of these compounds on fibrogenic genes. The mRNA levels of *ACTA2* (**A**), *MMP2* (**B**), and *TGFB1* (**C**) in LX-2 cells after treatment with the indicated compounds (10 μM), together with TGF-β1 (2 ng/mL) for 24 h were examined by real-time PCR (RT-PCR). The mRNA level was normalized against *GAPDH*. Data were presented as the mean ± SEM, ^#^
*p* < 0.05, ^##^
*p* < 0.01 as compared to that of control group; * *p* < 0.05, ** *p* < 0.01 as compared to that of TGF-β1 group. (**D**) The protein expression levels of α-SMA, MMP2, and TGF-β1 in LX-2 cells after treatment with the indicated compounds, together with TGF-β1 (2 ng/mL) for 24 h were detected by western blot assay. GAPDH served as internal reference. (**E**–**G**) Quantification of the intensities of α-SMA (**E**), MMP2 (**F**), and TGF-β1 (**G**) of (**D**). Data were presented as the mean ± SEM, ^#^
*p* < 0.05, ^##^
*p* < 0.01 as compared to that of control group; * *p* < 0.05, ** *p* < 0.01 as compared to that of TGF-β1 group.

**Figure 5 molecules-25-00773-f005:**
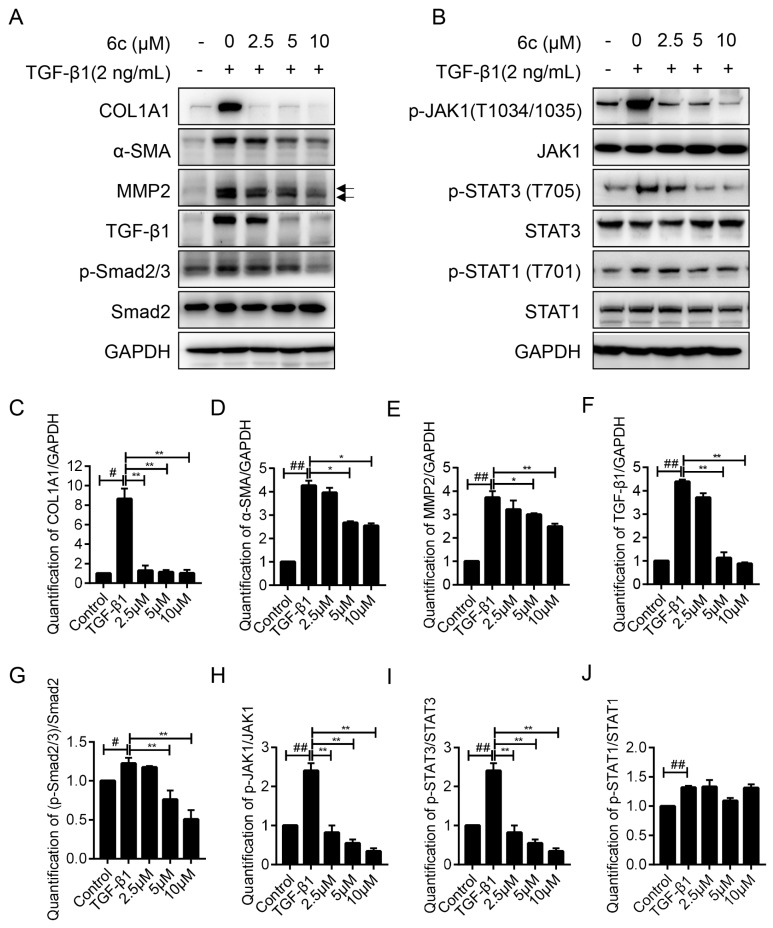
Inhibition effects of compound **6c** (0 μM, 2.5 μM, 5 μM, and 10 μM) on fibrogenic markers, TGF-β1/Smads pathway and JAK1/STAT3 pathway. (**A**) The protein expression levels of COL1A1, α-SMA, MMP2, TGF-β1, *p*-Smad2/3, and Smad2/3 in LX-2 cells after treatment with the indicated concentrations of **6c** together with TGF-β1 (2 ng/mL) for 24 h were detected by western blot assay. (**B**) The protein expression levels of p-JAK1, JAK1, p-STAT3, STAT3, p-STAT1, STAT1 in LX-2 cells after treatment with the indicated concentrations of **6c** together with TGF-β1(2 ng/mL) for 24 h were detected by western blot assay. GAPDH served as internal reference. (**C**–**G**) Quantification of the intensities of COL1A1 (**C**), α-SMA (**D**), MMP2 (**E**), and TGF-β1 (**F**) of (A). (**H–J**) Quantification of the intensities of *p*-JAK1 (**H**), *p*-STAT3 (**I**), and *p*-STAT1 (**J**) of (**B**). Data were presented as the mean ± SEM, ^#^
*p* < 0.05, ^##^
*p* < 0.01 as compared to that of control group; * *p* < 0.05, ** *p* < 0.01 as compared to that of TGF-β1 group.

**Figure 6 molecules-25-00773-f006:**
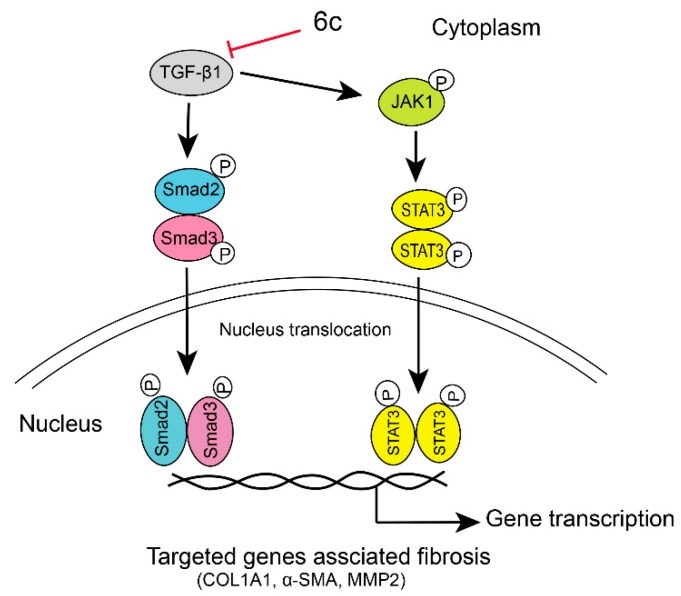
**6c** repressed liver fibrogenesis via repressing TGF-β1/Smads and JAK1/STAT3 pathways.

**Table 1 molecules-25-00773-t001:**
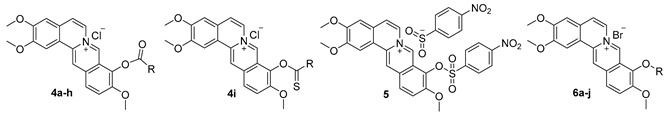
Structures and Inhibition effects on COL1A1 of target compounds.

Code	R	Inhibition Rate *^a^*	CC_50_ *^b^*	IC_50_ *^c^*	SI *^d^*
**1**	-	24.73 ± 3.27%	NT*^e^*	NT	NT
**4a**	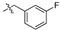	−5.12 ± 0.86%	NT	NT	NT
**4b**	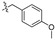	30.57 ± 7.77%	NT	NT	NT
**4c**	CH_2_CH_3_	13.12 ± 2.38%	NT	NT	NT
**4d**	(CH_2_)_2_CH_3_	6.42 ± 3.63%	NT	NT	NT
**4e**	(CH_2_)_3_CH_3_	16.05 ± 4.09%	NT	NT	NT
**4f**	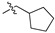	7.14 ± 1.79%	NT	NT	NT
**4g**	N(CH_3_)_2_	14.83 ± 1.42%	NT	NT	NT
**4h**	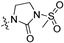	2.86 ± 2.69%	NT	NT	NT
**4i**	N(CH_3_)_2_	5.67 ± 4.61%	NT	NT	NT
**5**	-	26.78 ± 3.61%	NT	NT	NT
**6a**	CH_2_C_6_H_5_	38.16 ± 3.81%	NT	NT	NT
**6b**	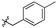	77.60 ± 2.93%	NT	NT	NT
**6c**	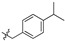	96.77 ± 5.64%	39.30 ± 2.09	3.98 ± 0.67	9.9
**6d**	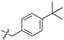	84.10 ± 7.91%	21.43 ± 3.98	5.56 ± 0.98	3.8
**6e**	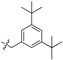	96.53 ± 3.00%	20.66 ± 1.54	4.47 ± 0.61	4.6
**6f**	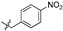	25.37 ± 4.40%	NT	NT	NT
**6g**	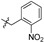	33.39 ± 3.71%	NT	NT	NT
**6h**	CH_2_CH_3_	26.19 ± 3.63%	NT	NT	NT
**6i**	(CH_2_)_3_CH_3_	26.74 ± 4.00%	NT	NT	NT
**6j**	CH_2_CH=CH	5.26 ± 1.78%	NT	NT	NT
EGCG	-	25.5 ± 7.90%	NT	NT	NT
DMSO	-	2.90 ± 0.00%	NT	NT	NT

*^a^* At the concentration of 20 μM; *^b^* Cytotoxic concentration required to inhibit LX-2 cell growth by 50%; *^c^* Half maximal inhibition concentration to inhibit the activity of COL1A1 promoter by 50%; *^d^* Selectivity index (CC_50_/IC_50_); *^e^* Not tested.

**Table 2 molecules-25-00773-t002:** Druglike Property Prediction of compound **6c**.

Code	Absn Risk *^a^*	CYP Risk *^b^*	TOX Risk *^c^*	ADMET Risk *^d^*
**6c**	0.994	1.8	1	3.825

*^a^* Druglike risk about absorption, suggested values: Absn Risk ≤ 3.5; *^b^* Druglike risk about metabolism, suggested values: CYP Risk ≤ 2.5; *^c^* Druglike risk about toxicity, suggested values: CYP Risk ≤ 2.0; *^d^* Druglike risk about ADMET, suggested values: ADMET Risk ≤ 7.5.

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
