# Peer review of "Discovery of 9O-Substituted Palmatine Derivatives as a New Class of Anti-COL1A1 Agents Via Repressing TGF-β1/Smads and JAK1/STAT3 Pathways"

_molecules, 2020, doi:10.3390/molecules25040773_

Round 1
Reviewer 1 Report
The manuscript is generally well written and evaluated a series of 9O-substituted palmatine derivatives for their inhibitory effect on COL1A1 promotor. The study indicated that 6c one main compound might anti-fibrogenic effect via repressing both canonical TGF-β1/Smads and non-canonical JAK1/STAT3 signaling pathways. The study is interesting and already underway, however, the present reviewer wants to shed light on some points in order to address this paper.
In fig 3, the protein expression levels of MMP2 and TGF-β1 in LX-2 cells after treatment with the indicated compounds, together with TGF-β1 for 24 h were detected by western blot assay. And also show the protein expression of MMP2 and TGF-β1 in LX-2 cells with compound 6c together with TGF-β1. The expression of protein were all with multiple bands. But why did they look different expression in figure 3D and 4A in the same cells and treated with same compound? Please check and indicate the main band(s). The authors made several protein expression on fibrogenic markers by western blot assay. Please quantification and statistical analysis the results, and give the samples number in the legends. In the study, the authors evaluate the safety profile of compound 6c in vivo. Could you show the data of the 6c effect, such as metabolic changes, survival rate, liver functions?Author Response
Q1: In fig 3, the protein expression levels of MMP2 and TGF-β1 in LX-2 cells after treatment with the indicated compounds, together with TGF-β1 for 24 h were detected by western blot assay. And also show the protein expression of MMP2 and TGF-β1 in LX-2 cells with compound 6c together with TGF-β1. The expressions of protein were all with multiple bands. But why did they look different expression in figure 3D and 4A in the same cells and treated with same compound? Please check and indicate the main band(s).
R1: Thanks very much for the comments. Due to the different run and exposure times in electrophoresis runs, the separations of the indicated proteins might look different. The expression of MMP2 in LX-2 cells was always with two bands, both of which were labeled by arrows, as seen in Figures 4D and 5A (originally 3D and 4A). At the meantime, the main band of TGF-β1 was labeled by arrow in Figures 4D (originally 3D). In addition, we rerun the western blot assay, replaced the band of TGF-β1 in Figure 5A (originally 4A).
Q2: The authors made several protein expressions on fibrogenic markers by western blot assay. Please quantification and statistical analysis the results, and give the samples number in the legends.
R2: Thanks for the advice. The expression of COL1A1 in Figure 2B was quantified by Image J and the statistical result was shown in Figure 3C (originally 2C). The expressions of a-SMA, MMP2 and TGF-β1 in Figure 4D (originally 3D) were quantified and shown in Figures 4E-4G. The expressions of COL1A1, a-SMA and other proteins in Figures 5A and 5B (originally 4A and 4B) were also quantified and shown in Figures 5C-5J.
Q3: In the study, the authors evaluate the safety profile of compound 6c in vivo. Could you show the data of the 6c effect, such as metabolic changes, survival rate, liver functions?
R3: Yes, the safety data of 6c were supplemented in the revised version, “All mice survived during the seven-day observation period, with glossy hair, fleshy body, agile movement, good appetite.” Unfortunately, the liver function index and pathological changes of the liver were not tested in the acute toxicity assay.
Reviewer 2 Report
This study screened the anti-COL1A1 Agents from 9O-substituted palmatine derivatives human hepatic stellate LX-2 cells. The results identified palmatine derivatives as a novel class of anti-fibrogenic agents.
Some comments:
1. In the introduction, the anti-HF background of palmatine (2) derivatives should be provided.
2. Use EGCG as the positive control of anti-COL1A1 Agent, but it should not a potent anti-fibrosis agent. Silymarin should suit for liver protection.
3. Western blot is a semi-quantification method, if possible, this study can provide the IC50 of 6c for inhibiting COL1A1 Protein expression using ELISA methods. In addition, the comparison between 6c and anti-fibrosis drug (ex. Sillymarin) is
4. Please discuss why 6c, 6d, 6e perform more anti-COL1A1 effect according to the chemical structure.
Author Response
Responses to Reviewer 2:
Q1: In the introduction, the anti-HF background of palmatine (2) derivatives should be provided.
R1: Thanks very much for the suggestion, and we had provided more detailed background as instructed.
Q2. Use EGCG as the positive control of anti-COL1A1 Agent, but it should not a potent anti-fibrosis agent. Silymarin should suit for liver protection.
R2: Thanks very much for the suggestion. Though EGCG is not a potent anti-fibrosis agent, its anti-fibrotic effect was reported by many groups. Our group identified that EGCG had anti-fibrotic effects in bile duct-ligated cholestatic rats [1], and Baba’s group reported that the supplementation of EGCG could effectively increase the mRNA expression of COL1A1 [2]. Therefore, we considered EGCG as a qualified positive control in our study. At the meantime, we totally agreed with your recommendation of silymarin, and we will try to apply it as a positive control in our future studies.
[1] Yu, D.K.; Zhang, C.X.; Zhao, S.S.; Zhang, S.H.; Zhang, H.; Cai, S.Y.; Shao, R.G.; He, H.W. The anti-fibrotic effects of epigallocatechin-3-gallate in bile duct-ligated cholestatic rats and human hepatic stellate LX-2 cells are mediated by the PI3K/Akt/Smad pathway. Acta Pharmacol. Sin. 2015, 36, 473.
[2] Kaida, K.; Honda, Y.; Hashimoto, Y.; Tanaka, M.; Baba, S. Application of Green Tea Catechin for Inducing the Osteogenic Differentiation of Human Dedifferentiated Fat Cells in Vitro. Int. J. Mol. Sci. 2015, 16, 27988.
Q3. Western blot is a semi-quantification method, if possible, this study can provide the IC50 of 6c for inhibiting COL1A1 Protein expression using ELISA methods. In addition, the comparison between 6c and anti-fibrosis drug (ex. Sillymarin)
R3: Thanks very much for the good suggestions, and we will take sillymarin as a new positive control in our future studies; on the other hand, we will also provide the IC50 of the key compounds for inhibiting COL1A1 expression using ELISA methods in the next manuscript in the near future.
Q4. Please discuss why 6c, 6d, 6e perform more anti-COL1A1 effect according to the chemical structure.
R4: Thanks very much for the suggestion, and we had added the discussion in the revised version.
Reviewer 3 Report
Brief Overview: Fan T et al. prepared and tested 20 90-substituted palmatine derivatives for suppression of various fibrogenic proteins and genes associated with hepatic fibrosis, in hepatic stellate cells (HSCs). Of the 20 derivatives, compound 6c resulted in the highest inhibitory effect against collagen α1 (COL1A1), α-smooth muscle actin (SMA), matrix metalloproteinase (MMP)-2, and transforming growth factor (TGF)-β possibly via the Janus kinase (JAK) 1/signal transducer and activator of transcription (STAT) 3 signaling pathway. The authors conclude that palmatine derivatives may be considered as a novel class of anti-fibrogenic agents.
Comments to the Authors:
This comprehensive work addresses an important health related problem. The manuscript is well written, and the methods are sound. The following changes are recommended:
Abstract: Acronyms should be spelled out at first mention. Introduction: While the rationale and objectives are clear, no working hypothesis is stated to indicate that the work was hypothesis-driven. Results and Discussion: Scheme 1 should be relabeled as Figure 2 to avoid confusion to the reader. All subsequent figures should be relabeled. Experimental: State the number of samples for each assay and the number of cells in each sample. For the toxicity study, explain why organ weights and histopathology were not assessed. Toxicology studies generally evaluate the potential of a substance (especially new compounds) to accumulate in a specific organ and/or tissue. This provides key information about the test compound such as absorption, absorption rate, distribution throughout the body, metabolism, route(s) of elimination, and dose effects on absorption, distribution, metabolism and elimination. Biological samples should include blood, urine, fat, muscles, liver, kidneys. Statistical Analyses: The analysis used, two-tailed Student’s t-test, is not appropriate for the design. This is a multi-compound (20 compounds), dose-response (4 doses) study. Student’s t-test is restricted to two groups. Since there are greater than two groups (Figures), one-way Analysis of Variance (ANOVA) should be used provided that the data are normally distributed, with Bonferroni correction. It is recommended that the authors consult a statistician.Author Response
Responses to Reviewer 3:
Q1. Abstract: Acronyms should be spelled out at first mention.
R1: Thanks for the suggestion, and we have revised the acronyms accordingly.
Q2. Introduction: While the rationale and objectives are clear, no working hypothesis is stated to indicate that the work was hypothesis-driven.
R2: Thanks for the informative comment, and we have revised the introduction accordingly.
Q3. Results and Discussion: Scheme 1 should be relabeled as Figure 2 to avoid confusion to the reader. All subsequent figures should be relabeled.
R3: Thanks for the suggestion, and we have revised as instructed.
Q4. Experimental: State the number of samples for each assay and the number of cells in each sample.
R4: We fully agree with the reviewer, and we have revised the number of samples and the number of cells in the revised version.
Q5. For the toxicity study, explain why organ weights and histopathology were not assessed. Toxicology studies generally evaluate the potential of a substance (especially new compounds) to accumulate in a specific organ and/or tissue. This provides key information about the test compound such as absorption, absorption rate, distribution throughout the body, metabolism, route(s) of elimination, and dose effects on absorption, distribution, metabolism and elimination. Biological samples should include blood, urine, fat, muscles, liver, kidney.
R5: Thanks for the constructive suggestion. We added the survival and behavior information in our revised version. The aim of this study is to initiate an anti-COL1A1 SAR of plamatine derivatives and assess their promise as anti-fibrogenic agents primarily. Yes, we agree that the organ weight change and histopathology behavior should be included in the assay of an ideal drug candidate, however, we believe there is still room for the activity improvement of these palmatine derivatives, and the SAR results presented in this study will help us to achieve more potent anti-fibrogenic agent(s) necessitating a systematic and comprehensive toxicity assay as well as pharmacodynamics study, which will be demonstrated in our future studies.
Q6. Statistical Analyses: The analysis used, two-tailed Student’s t-test, is not appropriate for the design. This is a multi-compound (20 compounds), dose-response (4 doses) study. Student’s t-test is restricted to two groups. Since there are greater than two groups (Figures), one-way Analysis of Variance (ANOVA) should be used provided that the data are normally distributed, with Bonferroni correction. It is recommended that the authors consult a statistician.
R6: Thanks for the fair criticism. In our revised version we improved the analysis. Bonferroni's multiple comparisons test of one-way Analysis of Variance (ANOVA) was applied instead to analysis the significance between groups. In Figures 5A and 5B (originally 4A and 4B), we designed five groups (control, TGF-β1, 2.5 μM, 5 μM, 10 μM) of the target compound of 6c. Every group was compared with the TGF-β1 group, and the adjusted P value is shown in figures 5C-5J.
Round 2
Reviewer 2 Report
The authors have revised the manuscript. This manuscript can be considered to accept.